

# Capturing temporal heterogeneity in soil nitrous oxide fluxes with a robust and low-cost automated chamber apparatus

Nathaniel C. Lawrence[1], Steven J. Hall[1]

[1] Department of Ecology, Evolution, and Organismal Biology, Iowa State University, Ames, IA, USA

*Correspondence to*: Steven J. Hall (stevenjh@iastate.edu)





**Abstract.** Soils play an important role in Earth's climate system through their regulation of trace greenhouse gases. Despite decades of soil gas flux measurements using manual chamber methods, limited temporal coverage has led to high uncertainty in flux magnitude and variability, particularly during peak emission events. Automated chamber measurement systems can collect high-frequency (sub-daily) measurements across various spatial scales but may be prohibitively expensive or incompatible with field conditions. Here we describe the construction and operational details for a robust, relatively inexpensive, and adaptable automated dynamic (steady-state) chamber measurement system modified from previously published methods, using relatively low-cost analyzers to measure nitrous oxide ($N_2O$) and carbon dioxide ($CO_2$). The system was robust to intermittent flooding of chambers, long tubing runs (> 100 m), operational temperature extremes (-12–39 °C), and was entirely powered by solar energy. Using data collected between 2017–2019 we tested the underlying principles of chamber operation and examined $N_2O$ diel variation and rain-pulse timing that would be difficult to characterize using infrequent manual measurements. Stable steady-state dynamics were achieved during the 29-minute chamber closure periods at relatively low flow rate (2 L min$^{-1}$). Instrument performance and calculated fluxes were minimally impacted by variation in air temperature and water vapor. Measurements between 08:00 and 12:00 were closest to the daily mean $N_2O$ and $CO_2$ emission. Afternoon fluxes (12:00–16:00) were 28% higher than the daily mean for $N_2O$ (4.04 versus 3.15 nmol m$^{-2}$ s$^{-1}$) and were 22% higher for $CO_2$ (4.38 versus 3.60 umol m$^{-2}$ s$^{-1}$). High rates of $N_2O$ emission are frequently observed after precipitation. Following four discrete rainfall events, we found an 12 to 26-hour delay before peak $N_2O$ flux, which would be difficult to capture with manual measurements. Our observation of substantial and variable diel trends and rapid but variable onset of high $N_2O$ emissions following rainfall support the need for high-frequency measurements.

## 1 Introduction

Soils play a critical role in Earth's carbon (C) and nitrogen (N) cycles. Managing soils to sequester C or reduce the emission of trace greenhouse gases $N_2O$ and methane ($CH_4$) is often suggested as an effective tool to combat climate change (Minasny et al., 2017; Paustian et al., 2016). Therefore, reliable trace gas measurements are critical for informing management. Although manual soil gas flux measurements have been collected for several decades, the high temporal and spatial variability of emissions has plagued attempts to obtain accurate and precise flux estimates needed to calculate annual budgets (Davidson et al., 2002; Groffman et al., 2009; Hutchinson and Mosier, 1981). Sampling at higher frequency than is practical with manual measurements may be required to constrain the role of soils in global biogeochemical cycles and validate the impacts of management practices on trace gas emissions (Barton et al., 2015; Merbold et al., 2015; Parkin, 2008). $N_2O$ emissions are particularly variable, so relatively less is known about peak emissions such as the time between rainfall and the subsequent $N_2O$ pulse that is frequently observed (Groffman et al., 2006, 2009). High frequency automated flux measurements that can span the large (>100 m) spatial scales that frequently accompany local topographical and hydrological variation may be critical to capture the dual spatial-temporal dynamics which are key to generating robust emission estimates.



Prefabricated automated chambers capable of measuring soil trace gas fluxes are available commercially and can be plumbed to a wide range of analyzers—most commonly, infrared gas analyzers that measure $CO_2$. Additional methods have been developed to measure other trace gases, including $N_2O$ and $CH_4$, by utilizing methods such as tunable diode laser or cavity ring-down spectroscopy technology. However, the high cost of the commercially available chambers and laser-based

analyzers, as well as their often-stringent operation and logistical requirements, put these methods out of reach for many field projects. We sought to implement a lower-cost soil gas flux measurement system capable of operating in a harsh field environment.

Environmental conditions, particularly those posed by flooding and agricultural management, created several unique challenges that could be expected in many field settings. Extreme heat and cold (-12 – 39 °C) and occasional submergence of

chambers mandated that our apparatus be tolerant of a wide range of conditions. Frequent agricultural management (tillage, planting, fertilization, harvest, etc.) at our field site required the chambers and associated equipment to be relatively portable so it could be removed to the field edge (~100 m away) and reinstalled several times per year (Fig. 1a). To avoid damaging crops, all equipment had to be movable on foot. Because electric power was unavailable, solar panels and batteries had to provide all necessary energy. Our core measurement system consisted of eight steady-state, flow-through chambers that each

quantified soil gas fluxes at each chamber every four hours. For one year, a second set of chambers was paired with the original eight for a total of sixteen chambers without sacrificing measurement frequency. With our design, chamber number and measurement frequency can be readily adjusted to fit study questions. We utilized two gas analyzers capable of measuring carbon dioxide ($CO_2$) and nitrous oxide ($N_2O$) concentration, respectively, although other gas analyzers could be employed with the chamber and manifold system described below. The gas analyzers were maintained in an instrument shed at the field

edge (Fig. 1a). This location was not impacted by flooding or agricultural management but was subjected to the temperature extremes noted above. We required analyzers that could operate effectively in these field conditions.

There is a rich literature on the impacts of chamber design and the potential biases of soil trace gas flux measurements. We chose a design that has been shown in field and laboratory experiments to provide accurate estimation of soil gas fluxes and isotopic composition (Bowling et al., 2015; Moyes et al., 2010a; Norman et al., 1997; Pumpanen et al., 2004). In one

comparison of different chambers, a variant of the open, flow-through design we used here measured known $CO_2$ fluxes produced in the laboratory to within 2–4 % of the actual values, which was relatively accurate compared to the other designs tested (Pumpanen et al., 2004). Pressure differential between the inside and outside of some chamber designs can create measurement artifacts (Fang and Moncrieff, 1998; Xu et al., 2006). The chambers described here utilize an open lid design (Fig. 2) that limits pressure differential to less than -0.2 Pa at the flow rate (2 L min$^{-1}$) we utilized (Rayment and Jarvis, 1997;

Moyes et al., 2010b). When using static chamber designs, soil gas flux is calculated as a function of the change in gas concentration over time within a closed chamber headspace. In contrast, with dynamic chambers we derive gas flux from the steady-state difference in concentration between air at the chamber inlet and pumped out of a chamber outlet (Fig. 2). When the outlet gas concentration is approximately constant, the chamber is at steady state. Steady-state chambers with low pressure differential have been shown to reproduce known $\delta^{13}C$ values of $CO_2$ fluxes (Moyes et al., 2010b), possibly because they have


less impact on the diffusive profile than many non-steady-state chamber designs (Nickerson and Risk, 2009). For our study, an additional consideration was that chambers needed to be located at variable distances (80–115-m) from the gas analyzers (Fig. 1a). We required this attribute to span a large (120 horizontal m) topographic gradient and to maintain analyzers and related instruments in a permanent location with vehicle access. As sampled gas can be vented downstream of the analyzers instead of routed back to the chamber (as is required for closed-loop static chamber designs), dynamic chambers can be located

at varying distances from the instruments without impacting the effective volume of the chamber headspace.

In this publication we present a method to construct a robust system of dynamic automated soil trace gas chambers along with the maintenance and troubleshooting lessons learned over the three-year period the chambers were running. In addition to presenting these operational details, we tested three underlying assumptions of our chamber design: (1) did chambers reach steady-state dynamics, (2) how did broad temperature fluctuations effect instrument performance in the field,

and (3) to what extent could water vapor impact our measurement values? We further utilized the high-frequency flux data to test two questions related to the temporal dynamics of gas emissions to inform manual sampling efforts: (4) how strong was the diel signal in trace gas emissions, and (5) what was the average delay between isolated rainfall events and the elevated $N_2O$ emissions that frequently followed.

## 2 Methods

### 2.1 Study Site


Our chambers were located at eight plots on 20 m intervals along a topographic gradient in a conventionally managed corn–soybean (*Zea mays–Gycine max*) agricultural field in central Iowa, USA (41.98° N, 93.69° W). The transect spanned 120 linear m (Fig. 1a), 2.25 m elevation, and included very poorly to moderately poorly drained soils (Mollisols classified as Okoboji to Clarion series under the USDA taxonomy). The lower half of the transect often experienced flooding after large rain events

(Logsdon and James, 2014) and chambers were occasionally completely inundated. The foreground of Fig. 1b shows one open and one closed chamber located in the lowest topographic position. The open chambers in the background are positioned along the topographic transect.

### 2.2 Chamber Design

The chambers we utilized were constructed in-house and various aspects were modified from previously published methods.

The chamber lid was first described by Rayment and Jarvis (1997), and Riggs and Stannard (2009) pioneered a pneumatic piston and stainless frame that opened and closed a chamber lid relative to a collar installed in the soil. Bowling et al. (2015) implemented a similar chamber design to measure $CO_2$ and $\delta^{13}C$ fluxes from a forest, but did not include extensive details on chamber design, construction, or operation.



The dimensions of many of the materials used were commercially specified with Imperial units but are reported here
in metric equivalents for consistency. Fig. 3 shows the chamber design. Chamber bases were constructed from 2.54 x 10.16
cm high-density polyethylene (HDPE) plastic (Fig. 3a). Custom L-brackets cut from 5.08 cm aluminum angle stock and bolted
to the plastic base provided two horizontal platforms to attach female spherical rod ends that served as the pivot point for
opening and closing the chamber (Fig. 3b). By routing vertical slots rather than drilling holes in the L-brackets, we provided a
means to adjust the lateral orientation of the pivot rod on each chamber after installation in the field (Fig. 3b). This was useful
to ensure that the chamber lid sealed against the collar given the inherent variability of soil microtopography. A 0.64 cm
diameter threaded rod between the rod ends provided an axle to attach the chamber frame (Fig. 3c). Most of the chamber frame
was constructed from 0.95 cm diameter stainless steel tubing; dimensions can be found in the caption and correspond to the
numbered labels in Fig. 3. To drill holes in the stainless tubing, we flattened the ends of each piece of tubing to a length of 1
cm in a bench vise and then drilled holes through the flattened portion to accommodate attachment bolts. The stainless tubing
was attached to the threaded rod described above or to aluminum angle brackets bolted to the chamber lid, noted by yellow or
red circles respectively in Fig. 3. Two lengths of 1.27 cm diameter stainless steel tubing surrounding a second 0.64 cm diameter
threaded rod and inserted into a 5.08 x 2.54-cm HDPE bar with a slot for a spherical rod end were attached to the end of a
pneumatic cylinder rod piston (Clippard, UDR-17-6) (Fig. 3d). Extension of the piston moved the chamber lid open or closed
and the HDPE bar and stainless tubing were used to prevent the threaded rod from flexing during movement of the chamber
lid. The three spherical rod ends, two located on the pivot point and one at the end of the cylinder piston served as rotational
degrees of motion (Fig. 3—yellow circles). All other connection points were rigid (Fig. 3—red circles).

The chamber lid followed a previous design which was shown to minimize the pressure differential between the inside
and outside of the chamber (< 0.2 Pa at flow rates of 4.5 L min$^{-1}$) (Moyes et al., 2010b; Rayment and Jarvis, 1997). The circular
chamber lid (38 cm diameter) was cut from HDPE panel (1.27 cm thick). A 2.54 cm diameter hole cut into the center of the
lid allowed a vertical gas inlet tube (Fig. 2a) to be fixed to the lid via custom-machined threads and a nut on the bottom of the
tube. The inlet tube (15 cm length) was machined from aluminum barstock and had internal and external diameters of 2.54
and 3.81 cm, respectively and a 2.54 cm length taper at the superior end (Fig. 2a). The inlet tube was covered by a polyvinyl
chloride (PVC) cap (10.16 cm diameter and 16 cm length; Fig. 2b) attached to the lid surface with three bolts, each with 1 cm
spacers to create an air gap between the cap and the lid surface (Fig. 2). The gap created by the spacers allowed atmospheric
air to flow to the inlet while preventing the direct horizontal flow of wind over the inlet tube opening. On the lower surface of
the lid, a D-shaped rubber seal (EPDM foam, 2.54 cm width) was affixed with silicone caulk in a ring where the lid contacted
the collar to create an air-tight seal when pressure was applied to the piston that closed the chamber (Fig. 1b). Early in our
study, we observed that high pressure (> 550 kPa) was needed to ensure a tight seal between the collar and chamber lid. To
minimize the piston air pressure required to seal the chamber lid against the collar, and thus conserve power, we bolted two
nested, 26 cm sections of slotted steel construction strut to the top of the chamber lid to provide additional mass (Fig. 3). Gas
from the inside of the chamber was sampled via a circular outlet manifold consisting of polyethylene tubing (6.4 mm OD, 3.2
mm ID) perforated by drilling 2 mm diameter holes through the tubing at 2 cm intervals, and was held in place approximately



3 cm below the lower surface of the lid with three stainless eyebolts. All tubing connections in our chamber and instrument manifolds were made using 0.64 cm brass Swagelok compression fittings. A threaded bulkhead union and tee fitting were used

to connect to the outlet manifold to external tubing above the chamber lid.

Chamber collars were made from PVC pipe segments (20 cm length, 30.48 cm ID) with the lower edge beveled with a belt sander to facilitate insertion into the soil. The beveled edge was pounded 10 cm into the soil for a total collar height of 10 cm and volume of approximately 7.3 L. The volume of air inside the longest length of tubing (120 m) connecting the chamber lid to the gas analyzers was < 1.8 L. To hold the chamber base in place relative to the collar, we initially used a ratchet

strap. However, we found that pressure exerted by the pneumatic arm when opening or closing the chamber occasionally shifted the position of the chamber base or collar and prevented a seal between the chamber lid, collar, and soil. This occasionally occurred following tillage or when soils were extremely dry. To address this problem, we anchored the chamber base using two steel rebar rods (60 cm length, 1.27 cm diameter) pounded 45 cm into the ground on either side of the chamber base and affixed to the outside of the chamber base with U-bolts positioned along the central axis of the collar (Fig. 3).


## 2.3 Chamber Lid Operation

Chambers were opened and sealed by alternatively applying 550 kPa pressurized air to either side of the pneumatic cylinder described above via two lengths of tubing connecting each chamber and the instrument shed (Fig. 4a). We used 0.64 cm OD, 0.43 cm ID low-density polyethylene (LDPE) plastic tubing. We initially used aluminum composite tubing (Synflex 1300),

which has been commonly used in other field trace gas measurement studies (e.g. Bowling et al. 2015), but we found this to be impractical for our application given its vulnerability to kinking during chamber installation and removal through dense vegetation. Pressurized gas tubing was connected to the pneumatic cylinder via National Pipe Thread (NPT) to Swagelock connections (Fig. 4a). Needle valves (Clippard JFC-2a) located between the pressurized tubing and either side of the pneumatic piston were used to manually adjust the rate of chamber opening and closing to prevent damage to the frame. Pressurized gas

was initially supplied by a pressurized cylinder and regulator as described in Riggs et al. (2009). However, we found that cylinders were impractical to supply the volume of gas necessary to pressurize the ~100 m lengths of tubing between the cylinder and chambers with frequent opening/closing. To provide a less labor-intensive source of pressurized air, we installed a Gast 12 VDC oil-less air compressor regulated by an air compressor switch (Condor MDR 3) with cut-in pressure set to 450 kPa and cut-out pressure set to 550 kPa (Fig. 4b). It was important to remove excess moisture from the pressurized air to

maintain downstream metal components and valves. A fifty-foot coil of copper tubing immediately downstream of the compressor allowed the pressurized air to cool and water to condense. Excess moisture was removed by a water trap (Speedaire #4ZL49) connected to an additional 1-liter reservoir made from PVC pipe and Swagelok fittings, which was periodically drained via a needle valve to the exterior of the instrument enclosure (Fig. 4c). From the water trap, the pressurized air flowed to a manifold of four-channel, two-way valves (Clippard MME-41PEEC-W012) which controlled the open/sealed position of





each chamber by suppling pressure to either of two lengths of tubing extending to each chamber (Fig. 4a,d). Each valve was wired to one channel of a 12 V datalogger-controlled relay controller (Campbell Scientific SDM-CD16AC) such that pressurized air maintained the chamber in an open position when the relay was closed.

**2.4 Principles of Chamber Gas Sampling**

Fig. 5 outlines the movement of sample gas between chamber and analyzers. Air was pulled through two separate tubes. One
tube sampled gas adjacent to the chamber inlet tube (Fig. 2), while the second pulled air from the perforated tubing manifold inside the chamber (Fig. 2). The second sampling tube is referred to here as the chamber outlet, as it served to pull ambient air from the chamber inlet tube through the chamber. Both sampling tubes were filtered through 1 μm Teflon (i.e., hydrophobic) filters (Pall Corporation) affixed to LDPE tubing via Swagelok connections immediately outside the chamber to prevent any particulates and liquid water from being pulled through the tubing (Fig. 5a).

200        Chamber sample selection was achieved by two sets of eight normally closed solenoid valves, one for inlet and one for outlet selection (Clippard, DV-2M-12-L, Fig. 5b). Downstream of the chamber selection manifolds, both inlet and outlet gases flowed through additional 1 μm filters. Inlet and outlet flow rates were set independently by two mass flow controllers (Aalborg, GFCS-010201) upstream of two 12V diaphragm gas pumps (KNF Neuberger UNMP830; Fig. 5c). Both flow rates were set to 2 L min$^{-1}$ by the mass flow controllers, and actual flow rates were recorded on the datalogger (which was important
for diagnosing potential problems during operation, as discussed later). To mediate selection of the gas sample that flowed to the analyzers, a third sample selection manifold with four normally closed solenoid valves selected between inlet, outlet, high concentration standard, and low concentration standard (Fig. 5d). To maintain a constant flow rate through the inlet and outlet sampling tubes when the sample was not being routed to the analyzers, two needle valves vented excess flow between the gas pumps and the selection manifold (Fig. 5). The selected sample gas flowed through a common sample gas mass flow controller
set to 0.9 L min$^{-1}$ (Fig. 5e). An internal pump in the N$_2$O analyzer sampled gas at 0.8 L min$^{-1}$, and this pump also served to pull sample through the CO$_2$ analyzer which had no internal pump. The remaining 0.1 L min$^{-1}$ was vented through a final needle valve placed upstream of the CO$_2$ analyzer (Fig. 5e).

Two instruments in series were used to analyze CO$_2$ and N$_2$O, respectively (Fig. 5e). The CO$_2$ analyzer was placed upstream of the N$_2$O analyzer to avoid artefacts from the high oven temperature in the latter. We used either a LI-COR 830
(or subsequently, LI-COR 850) Infrared Gas Analyzer to measure CO$_2$ concentrations by infrared absorbance. Downstream, a Teledyne 320U gas filter correlation analyzer measured N$_2$O concentration via infrared absorbance by frequently comparing the sample to a reference gas in a rotating filter (Fassbinder et al., 2013). Instantaneous gas concentrations, as well as the air temperature, inlet flow, outlet flow, and sample flow were measured every 10 seconds and recorded on a datalogger (Campbell CR3000).



## 2.5 Measurement Principle

Each chamber flux measurement was conducted over the course of a half-hour cycle. At the beginning of each half-hour cycle when a new chamber was going to be measured, a chamber lid was closed by triggering a relay to apply pneumatic pressure to the piston, and the inlet and outlet sampling tubes of the respective chamber began to be sampled at 2 L min⁻¹. Both inlet and outlet tubes were sampled continuously at a constant rate during the half-hour cycle while a downstream selection manifold alternated which gas was routed to the instruments with residual flow vented to the instrument shed through needle valves (Fig. 5). All pneumatic and sample selection valves were controlled by the datalogger. Calibration gases (standards) were measured every two hours (Fig. 5d). If standards were measured during a given chamber measurement sequence, this was conducted at the beginning of the half-hour period: each standard was measured for three minutes by opening a valve on the gas selection manifold while chamber inlet and outlet flows were vented (Fig. 6a,b). During measurement periods where standards were not measured, the inlet sample was opened first on the selection manifold (Fig. 6c). After 11 minutes, the inlet was vented while the outlet sample was routed to the instruments until the 16th minute of the half hour (Fig. 6d). The first inlet and outlet gas concentration values from a given chamber measurement cycle (Fig. 6c and 7d respectively) were not used to calculate fluxes, as the chamber headspace concentrations of $CO_2$ and $N_2O$ were often not at steady state during this time. These values, however, were useful for troubleshooting and assessing temporal trends in chamber gas concentrations. Between minutes 16–21 and 21–29, the inlet and outlet were respectively measured for a second time (Fig. 6e,f). The minimum 5-minute measurement period for inlet and outlet samples was chosen to overcome a lagged response in the $N_2O$ analyzer following a switch in sample gas composition, which was as long as two minutes when there were large concentration differences between inlet and outlet samples; the $CO_2$ analyzer typically stabilized much faster (tens of seconds). The difference between the inlet and outlet gas concentrations averaged over the last two minutes of their second respective measurement period (Fig. 6e,f) were used to calculate soil gas fluxes (units of umol m⁻² s⁻¹) using Eq. (1). The last 10 seconds of data from each period were excluded because of transient values during valve switching.

$$Flux = \frac{(PF)(ConcOut - ConcIn)}{(R*T*A)} \tag{1}$$

Where P is equal to mean atmospheric pressure at our study site (atm), F is outflow rate (L s⁻¹), ConcOut is the standard-corrected second outlet measurement period gas concentration (umol mol⁻¹) (Fig. 6f), ConcIn is the standard-corrected second inlet gas concentration (umol mol⁻¹) (Fig. 6e), R is the ideal gas constant (L atm K⁻¹ mol⁻¹), T is temperature (K), and A is the area covered by the chamber (m²). Following the end of the measurement period (29 min total), the chamber was opened by applying pneumatic pressure to the opposite end of the piston via the open/sealed manifold (Fig. 4a, d) and would remain open prior to the next measurement sequence.

Corrected gas concentration values were obtained by applying two-point linear standard corrections updated every two hours (e.g. Fig. 6a,b). The instrument output during the last minute of each standard measurement, again excluding the last 10 seconds, was averaged for calibration. Corrected gas concentrations were obtained by regressing measured standard values against known values to obtain a linear slope and intercept used to correct raw values. Working standards were prepared

by filling two 50-L gas cylinders with higher and lower concentrations of analytes by mixing $CO_2$- and $N_2O$-free air (zero air) with a concentrated standard gas to achieve values that approximately spanned the range of $CO_2$ and $N_2O$ concentrations

observed in the field. The mole fractions of each standard gas were verified by analyzing five replicates each on a gas chromatograph (Shimadzu 2014A) with thermal conductivity and electron capture detectors, which were calibrated according to additional NIST-traceable standards using a four-point curve. Gas cylinders filled to 140 MPa lasted approximately nine months.

## 2.6 Power Supply: Solar Panel/Batteries

At our field site, six 265-W solar panels (Kyocera) with 16 deep cell marine batteries (Trojan J305E-AC 6V) were able to power the analysis system for much of the year. Figure 7 illustrates the solar charging and battery storage system. Two sets of three solar panels each were wired in series through parallel 15-amp circuit breakers within a combiner box. The positive lead flows through a 30-amp circuit breaker with a second combiner box before joining the negative at a charge controller (Morningstar TS-MPPT-60, Fig. 7). Indicator lights on the charge controller were used to assess the remaining battery charge,

and we occasionally shut the entire system down during prolonged periods of low sunlight to avoid completely discharging the batteries. The charge controller positive output flowed through a 63-amp circuit breaker (Fig. 7) to the final positive lead of a battery bank consisting of four sets of four serially wired batteries, each connected in parallel (Fig. 7). The negative output from the charge controller flowed to the negative lead at the opposite end of the battery bank. A 24 VDC output connected to a 60-amp breaker (Fig. 7) and a DC/AC converter provided power for the 110 VAC $N_2O$ analyzer. A subset of two batteries

provided 12 VDC power to the other components (datalogger, $CO_2$ analyzer, switches, valves, and additional sensors).

## 3 Results and Discussion

### 3.1 Troubleshooting

While often no maintenance was required, we typically checked the measurement system every several days to prevent data gaps if a failure occurred. Under ideal conditions (permanent chamber installation, ample sunlight, no flooding), the analysis

system may be able to operate over periods of weeks to months without maintenance. However, we found that problems related to chamber submergence, component failure, or unintended faunal interactions occurred on occasion. This section highlights some common issues and practices that we found helpful for addressing them.

### 3.1.1 Excess Moisture

Periodic flooding presented one of the greatest challenges at our field site. Chambers could not sample gas when the water

level was above the height of the perforated outlet manifold suspended from the chamber lid (~7 cm above the soil surface). When water exceeded this height, the filter located at the chamber outlet (Fig. 5a) became saturated with water and stopped



flow, preventing damage to the downstream components. If flooding exceeded the height of the inlet (~30 cm depth), the inlet filter was similarly impacted. Data affected by saturated filters was flagged by noting below-normal inlet/outlet flows during post-processing and was removed. We replaced saturated filters after the water level receded to return the chamber to operation.

Wet filters were dried at 100°C and reused. Excess water also created problems when it condensed downstream of the air compressor. During humid summer conditions the compressor water trap reservoir (Fig. 4c) was emptied at least once every two weeks. In sub-freezing conditions the trap rarely collected water but was emptied after warmer periods to prevent expansive bursting when temperatures returned below 0°C. Pumps and valves occasionally failed for unknown reasons. In general, we identified problems related to gas flow and sample selection by plotting flow rates over time for each chamber

measurement sequence during data post-processing and replaced any faulty components.

### 3.1.2 Gnawing Animals

Early in our experiment, animals occasionally chewed through the gas tubing between the instrument shed and the chambers. For protection and organization, all four tubes connecting each chamber to the instrument shed (comprising chamber inlet and outlet gas samples, and compressed air for opening and closing the chamber, respectively) were subsequently wrapped in 2.54-

cm diameter polyethylene split corrugated wire loom tubing (Drossbach 25D260). The last several cm of each of the four tubes must be able to move independently to allow the piston to move and the chamber lid to open and close. To protect these final portions of tubing which could not be wrapped in protective loom tubing, we replaced the last 30 cm of tubing with semi-flexible 0.64-cm diameter copper tubing connected with Swagelok fittings. The copper tubing was molded to enable necessary movement of chamber components and was not impacted by animals. We documented and isolated leaks by capping the

chamber end of each tubing line, applying pressure with an air tank to each individual tube, and checking for a drop in regulator pressure. Large leaks were audible and could be easily found and repaired by splicing in replacement tubing using Swagelok union fittings. To test for small leaks, we plumbed the valves to a tank of industrial-grade helium and used a helium-specific leak detector (Restek 28500). After protecting against animal damage, leaks were infrequent.

### 3.1.3 Power Limitation

We experienced occasional power outages during extended periods of cloudy weather and during winter. By periodically turning the analysis system off for several days to allow the batteries to reach full charge, we could collect 2–3 days of measurements even in cold/cloudy conditions. The Teledyne $N_2O$ analyzer has an internal component (heated to near 70°C) which consumed additional power during cold weather. We found that enclosing the $N_2O$ analyzer in a plywood box with 2.54 cm polystyrene foam insulation on four sides (leaving one side and the back open for ventilation) reduced power use. We also

adjusted the angle of the solar array at least twice a year to increase efficiency. Collectively, these energy efficient measures allowed the instrument to operate for longer periods when solar energy was limiting. Occasionally, however, the DC/AC converter would shut down during the night due to power limitation and would turn on again when sunlight was available.



Data from the $N_2O$ analyzer were consistently biased during an 8-hour period as the instrument warmed up. We flagged and discarded these data during post-processing by plotting analyzer output over time and removing peaks following periods where
no output was recorded.

## 3.2 Measurement Assumptions

A key principle of steady-state chamber operation is that the gas concentration inside the chamber headspace is approximately at equilibrium (gas flux from the soil is balanced with gas removed via the chamber outlet) when the flux measurement is made. The time to achieve steady-state conditions is a balance between the soil flux rate and the flow of gas through the
chamber. Here, to enable the use of smaller pumps and conserve power we employed lower flow rates (2 L min$^{-1}$) than often employed previously in dynamic chambers (e.g. 4 L min$^{-1}$; Bowling et al., 2015). Initial tests revealed that use of larger pumps needed to achieve 4 L min$^{-1}$ flow rates over > 100 m tubing runs was not sustainable from the perspective of power supply. To validate the steady-state assumption at 2 L min$^{-1}$, we analyzed the slope of a linear regression between concentration of $CO_2$ and $N_2O$ and time over the final outlet measurement period (Fig. 6f, approximately minutes 27–29) using data from three
separate periods chosen to cover a broad range of fluxes and spanning two weeks in total. We found an average increase of $0.18 \pm 10.51$ ppm $CO_2$ min$^{-1}$ (mean and SD) and $0.57 \pm 8.40$ ppb $N_2O$ min$^{-1}$, respectively, indicating that both gases were approximately at steady state at the end of the measurement period (relative to mean chamber outlet values of 684 ppm and 494 ppb for $CO_2$ and $N_2O$, respectively). We repeated this analysis for the final inlet measurement period (Fig. 6e, approximately minutes 19–21) and found a change of less than one ppm or ppb min$^{-1}$ $CO_2$ and $N_2O$ relative to mean chamber
inlet concentrations of 539 ppm and 331 ppb, respectively.

To assess temperature sensitivity of both gas analyzers under field conditions we examined the slope and intercept of standard curves measured during a 20-day period when air temperature ranged from -4 °C to 21°C and during which the instruments ran continuously. There was no significant directional trend in air temperature over this period to avoid conflating temperature-related drift and drift of the instrument over time unrelated to temperature. All four metrics examined (slope and
intercept of $CO_2$ and $N_2O$ calibration curves) displayed correlations with temperature. However, the impact of temperature on the slope of the $CO_2$ and $N_2O$ calibrations was less than $10^{-3}$ ppm °C$^{-1}$ for both values. These values correspond to less than 1% difference in instrument output between the highest and lowest temperature near ambient gas concentration for $CO_2$ and $N_2O$ (400 and 0.3 ppm respectively). The intercept values showed greater sensitivity (0.02 and 0.003 ppm °C$^{-1}$ for $CO_2$ and $N_2O$ respectively). These values correspond to approximately 0.5 ppm difference in $CO_2$ and 0.08 ppm difference in $N_2O$ at
the high and low temperature range observed. Taken together, we found that the $N_2O$ instrument had a -0.006 ppm °C$^{-1}$ sensitivity, in close agreement to the -0.009 ppm °C$^{-1}$ found by Fassbinder et al. (2013). As detailed above, standards were measured every two hours to account for instrument sensitivity to environmental conditions. Additionally, because gas flux was calculated as the difference between and inlet and outlet concentration the intercept values cancelled mathematically,





thereby removing any additional bias due to temperature-related intercept drift between standard measurements. Therefore,
temperature variation between measurements had negligible impact on the final flux calculation.

Optical trace gas measurements may be affected by a number of interacting factors including temperature, pressure, and water vapor pressure (McDermitt et al., 1993). Water vapor can be removed through chemical traps. However, the high gas flow in our system (2 L min⁻¹) made reagent replacement in chemical traps impractical, and preliminary work showed that membrane-based driers did not always completely remove water vapor in our operating environment, where relative humidity
often reached 100%. The $N_2O$ analyzer we utilized removed moisture through a multi-tube Nafion dryer (Model NMP850KNDCB, KNF Neuberger Inc.). Water vapor was not removed prior to measuring $CO_2$ concentration. As we calculated the soil $CO_2$ flux as proportional to the concentration difference between inlet and outlet gases, we were primarily concerned with a change in water vapor between the inlet and outlet measurement (Fig. 6e, f). In 2019, measurements were made with a LI-COR 850 that included a water vapor correction and measurement, which we used to constrain the potential
impact of water vapor on our previous $CO_2$ measurements. McDermitt et al. (1993) found that the required water vapor correction using a similar analysis was < 10 ppm $CO_2$ at water vapor pressure of 25.3 mmol mole⁻¹ and $CO_2$ concentration up to 1000 ppm. Water vapor pressure in the gases we measured spanned 1.0–53.6 mmol mole⁻¹, with an average difference between inlet and outlet gas of 1.8 mmol mole⁻¹ and a maximum of 36.4 mmol mole⁻¹. These small observed changes in water vapor between inlet and outlet measurements indicate a minor impact on measured $CO_2$ fluxes: if the water vapor difference
between inlet and outlet caused a <10 ppm bias in the measured $CO_2$ concentration (as expected in >99.9% of our observations), this would impact the average measured $CO_2$ flux (3.47 umol m⁻² s⁻¹) by <5.2% (0.18 umol m⁻² s⁻¹), which is within the typical range of measurement uncertainty for reproducing a known flux value under controlled conditions (Pumpanen et al., 2004). The correction under a more moderate water vapor difference between inlet and outlet (<12.6 millimoles mole⁻¹) that spans >97% of observed differences is approximately half the impact of this extreme example (0.09 umol m⁻² s⁻¹). Unrelated to its
impacts on instrument performance, water vapor can also impact flux measurements by dilution (Harazono et al., 2015). Given an average water vapor difference between inlet and outlet of 1.8 mmol mole⁻¹ and maximum of 36.4 mmol mole⁻¹, impacts of dilution on measured fluxes would also be small: typically <0.18% and as much as 3.6%.

To constrain the potential impacts of water vapor on measured $N_2O$ concentrations, we conducted a simple laboratory experiment comparing the $N_2O$ instrument output between a high and low moisture measurement on a three-point standard
curve. Water vapor was assessed with a LI-COR 850 installed in-line and upstream of the $N_2O$ sensor. To quantify the impact of water vapor on instrument output, we compared the standard curve created from dry standards to a curve created after bubbling the gas through a jar of deionized water. The bubbling technique added on average 25.4 millimoles mole⁻¹, spanning >99.9% of observations of the difference between water vapor at the inlet and outlet in the field. Standard gases ranged up to 9.96 ppm $N_2O$, greater than all differences between inlet and outlet observed in the field. No difference was noted in $N_2O$
instrument output due to the presence of water vapor, which suggested the drying column was effective at removing water vapor or that the gas filter correlation method corrected for any impacts of residual vapor.



### 3.3 Temporal Dynamics

Manual sampling by field crews is generally accomplished during normal daytime work hours. In contrast, automated measurements can be scheduled throughout the 24-hour diel period. Figure 8 displays boxplots of $N_2O$ emission from days
when chambers were measured at each four-hour interval during 2017 and 2019 (the years of *Zea mays* cultivation). Though infrequent, we observed occasional instantaneous negative $N_2O$ flux values, as observed in other ecosystems including in cultivated soils (Schlesinger, 2013; Wu et al., 2013). Figure 9 shows the $N_2O$ and $CO_2$ emissions from two typical one-week periods from September 2017 and August 2018. A diel trend is visible for most chambers in August and some chambers and time periods in September. In general agreement with previously published auto-chamber $N_2O$ studies from agricultural soils,
we found the lowest rates of emission during early morning (04:00–08:00) and highest emissions during early afternoon (12:00–16:00) (Akiyama et al., 2000; Alves et al., 2012; Bai et al., 2019; Flessa et al., 2002; Savage et al., 2014). Estimates created for each day and chamber using early afternoon measurements were on average 28% greater than the daily average from each chamber (4.04 versus 3.15 nmol $m^{-2}$ $s^{-1}$) and this difference varied from -13.9 to 110 nmol $m^{-2}$ $s^{-1}$. The relative difference between average and peak daily emissions was in reasonable agreement with previous data from agricultural fields
in the United Kingdom, Australia, and the United States (approximately 31, 47, and 33% respectively [Alves et al., 2012; Bai et al., 2019; Savage et al., 2014]). Although $CO_2$ emissions displayed highest and lowest fluxes during the same time periods as $N_2O$, the early afternoon $CO_2$ values averaged only 22% greater emissions than the daily average (4.38 and 3.60 umol $m^{-2}$ $s^{-1}$), and this difference varied between -7.72 and 21.1 umol $m^{-2}$ $s^{-1}$.

$N_2O$ emissions pulses have often been observed following rain events (Savage et al., 2014; Sehy et al., 2003). To
assess the length of the delay between rainfall and peak emissions, we analyzed the number of hours between heavy rainfall (>2 cm total over 24 h) and subsequent peak $N_2O$ emission rate averaged over all chambers. A rain gauge located on-site recorded precipitation data that was collected and obtained through the Iowa Flood Information System (IFIS, 2017). There were 45 days with total rainfall over 2 cm. To avoid conflating more than one rain event, we chose isolated events without rainfall in excess of 4 mm $d^{-1}$ in the preceding or the following two days. Of the 15 isolated rain events observed, four were
analyzed that did not span data gaps (Fig. 10). The rain to peak emission delay varied from 12 to 26 hours among precipitation events which varied from 2.4 to 4.4 cm.

### 4 Conclusions

Our results indicate that steady-state conditions were achieved under reasonable periods of chamber closure (29 min; equivalent to the common 30-min averaging interval for eddy covariance measurements [Loescher et al., 2006]) and flow rates
(2 L $min^{-1}$) that could be attained using low-power 12V pumps. The results were minimally impacted by measurement error due to water vapor and were robust to changes in temperature. We applied our high frequency data to address two questions, how strong does diel variation impact trace gas emissions and how long is the delay between precipitation and the frequently observed pulse in $N_2O$. Our observations showed that the average daily emissions were most closely approximated by



measurements made between 08:00 and 12:00. Though $CO_2$ emissions were best approximated during the same time interval,
the difference between peak emissions and the daily average was less pronounced and displayed less variability. We found the
delay between rainfall and peak $N_2O$ emissions varied between 12 and 26 hours, intervals that would be difficult to capture
using manual sampling methods. Both findings of temporal variability support the need for high-frequency measurements to
calculate annual soil trace-gas emissions budgets. This measurement system could also be adapted to study other gases
provided the gas analyzers chosen are able to tolerate field conditions. In particular, the steady-state chamber design used here
provides a powerful tool for future studies to couple gas flux with isotopic measurements that may uncover the source and
processes underlying the observed flux.

Agricultural management required us to remove the chambers and associated equipment several times of year, by
hand. Without these constraints, experiments utilizing this method could examine processes that take place on even greater
spatial scales than those utilized here (tubing runs > 100 m) and with a greater number of chambers. Despite these challenges,
we were able to construct and maintain 16 high-frequency automated chambers for sub-daily $N_2O$ and $CO_2$ flux measurements
in a temperate agricultural field, with a total materials cost (~$40,000 US dollars, including parts for chambers, gas analyzers,
control system, and power supply) that is a fraction of the cost of most optical $N_2O$ analyzers alone.

**Code/data availability**

Raw data and post-processing scripts are available from the corresponding author on request.

**Author Contribution**

N. C. L. and S. J. H. jointly designed and carried out research and prepared the manuscript.

**Competing Interests**

The authors declare that they have no conflict of interest.

**Acknowledgements**

This work was funded in part by USDA NIFA (award 2018-67019-27886), the Leopold Center for Sustainable Agriculture
(award E2017-02), the Iowa Nutrient Research Center (award 109-47-03-39-3650), and Iowa State University faculty startup
funds. We thank Carlos Tenesaca, Anthony Mirabito, Lucio Reyes, and Lindsay Mack for field assistance.


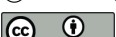



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


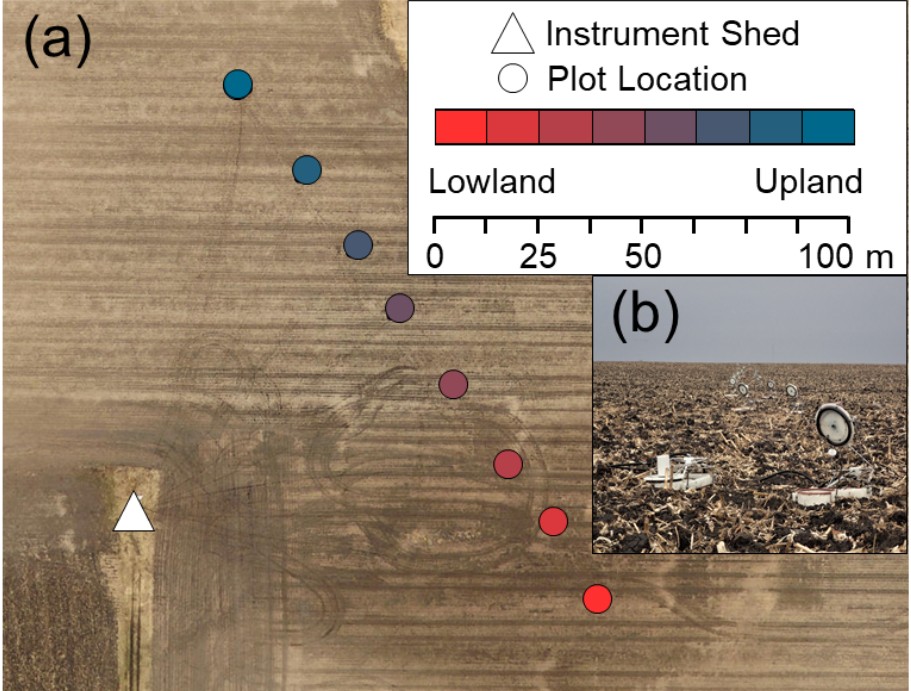


**Fig. 1a Aerial image of the field site with plot locations. 1b Image from the lowest topographic position along the transect. Front left: a closed chamber. Front right: an open chamber between measurements. The transect is visible in the background. Image Source: Esri, DigitalGlobe, GeoEye, Earthstar Geographics, CNES/Airbus DS, USDA, USGS, AeroGRID, IGN, and the GIS User Community.**






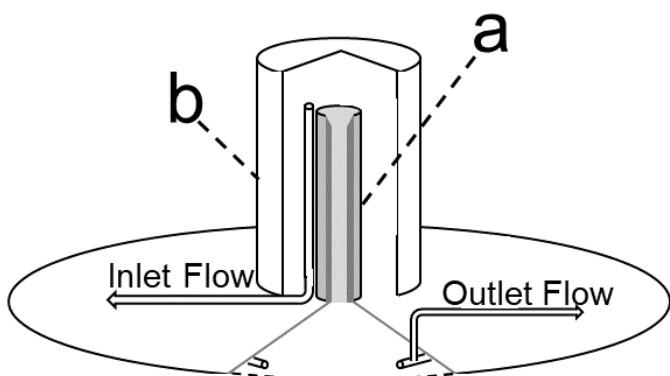


**Fig. 2. Illustration modified from Rayment and Jarvis (1997) depicting the chamber lid with cut-out to show the inlet tube a and the PVC cap b. Inlet and outlet sampling points are noted.**





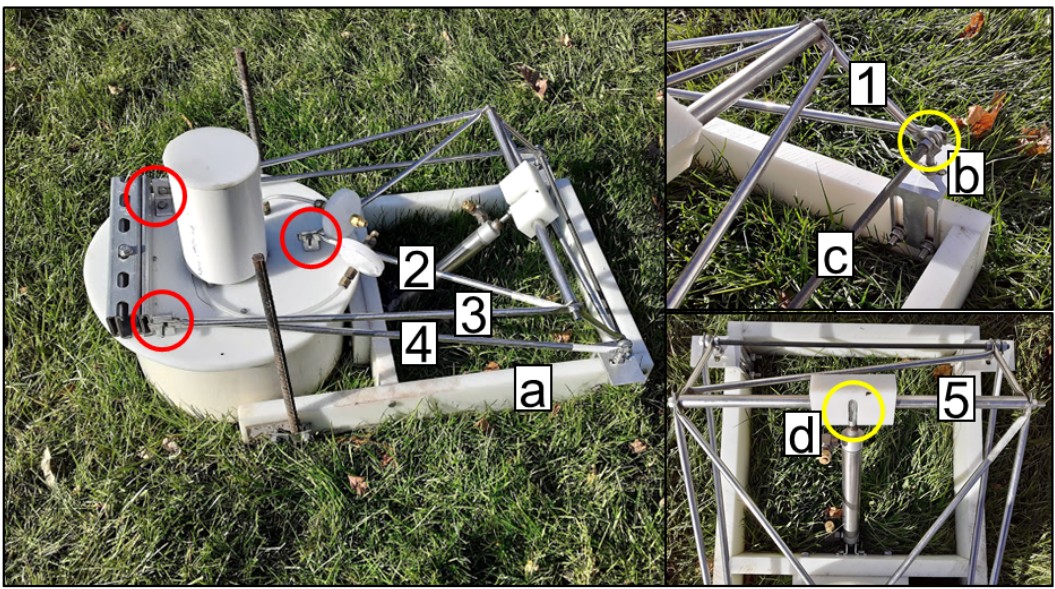

**Fig. 3. Image of chamber design: HDPE base a, aluminum L bracket b, threaded rod c, pneumatic cylinder rod end d. The length of each numbered stainless steel tube is as follows: 1 (16 cm), 2 (41 cm), 3 (56 cm), 4 (65 cm), 5 (18 cm). The yellow circles indicate**

**rotational degrees of motion while red circles denote rigid, fixed connection points.**





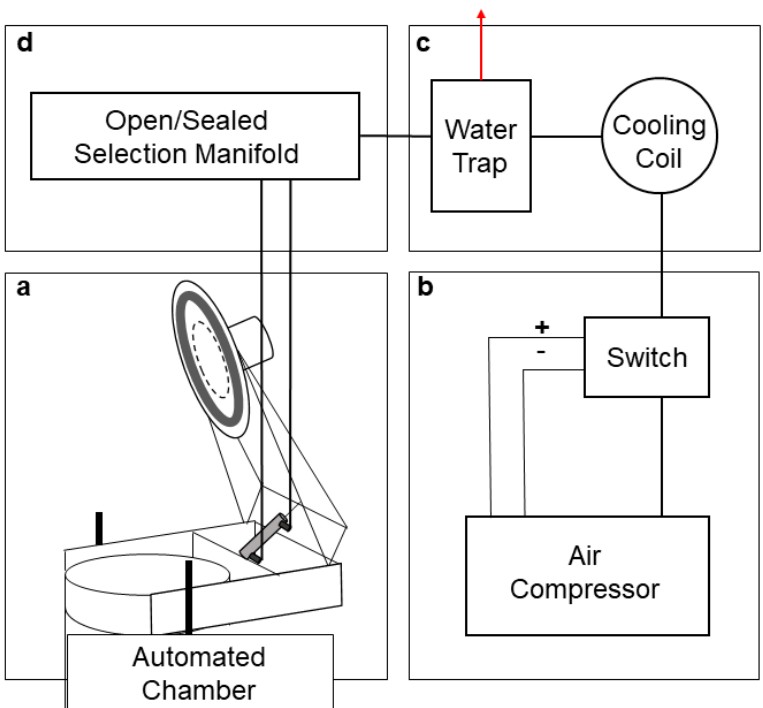

Fig. 4. Illustration of the chamber pneumatic system that controls opening and closing of chambers. The red arrow denotes the tube to drain the water trap reservoir.






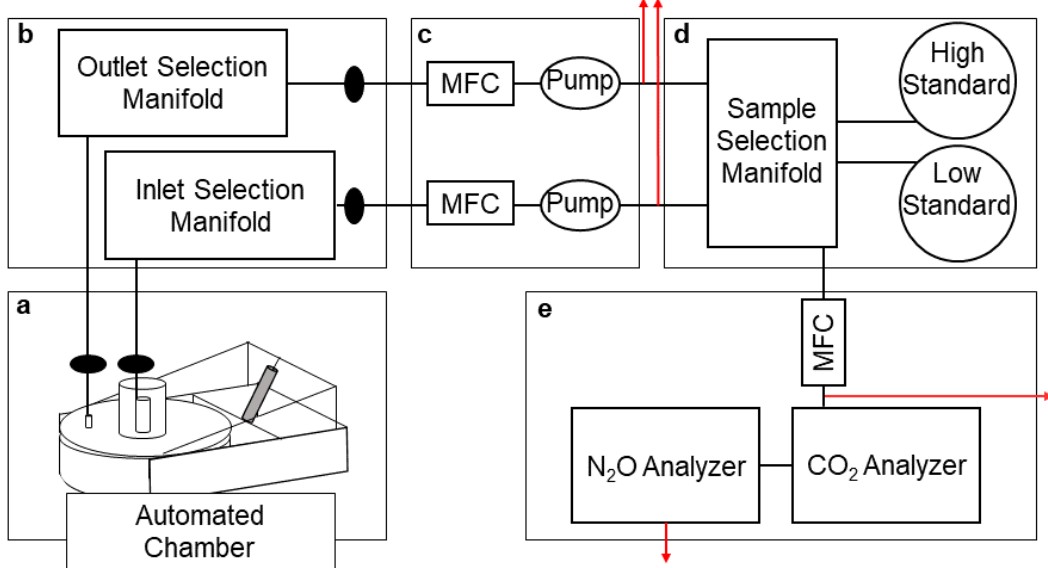

**Fig. 5. Schematic of the sample selection system. Mass flow controllers are abbreviated MFC. Filters are denoted by black ovals.**
**Red arrows indicate where needle valves vent excess flow.**




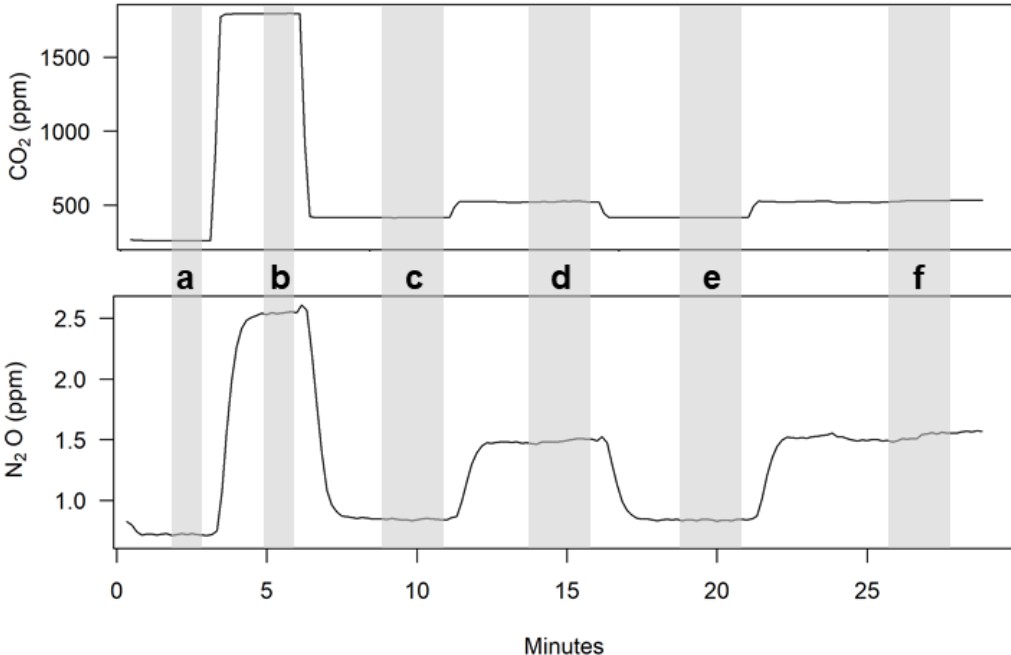


**Fig. 6. Raw instrument output over a representative half-hour chamber measurement period: low standard a, high standard b, first inlet measurement c, first outlet measurement d, second inlet measurement e, second outlet measurement f. The second set of inlet/outlet measurements was used for flux calculations. Shaded bars indicate periods where output was averaged for subsequent calculations.**




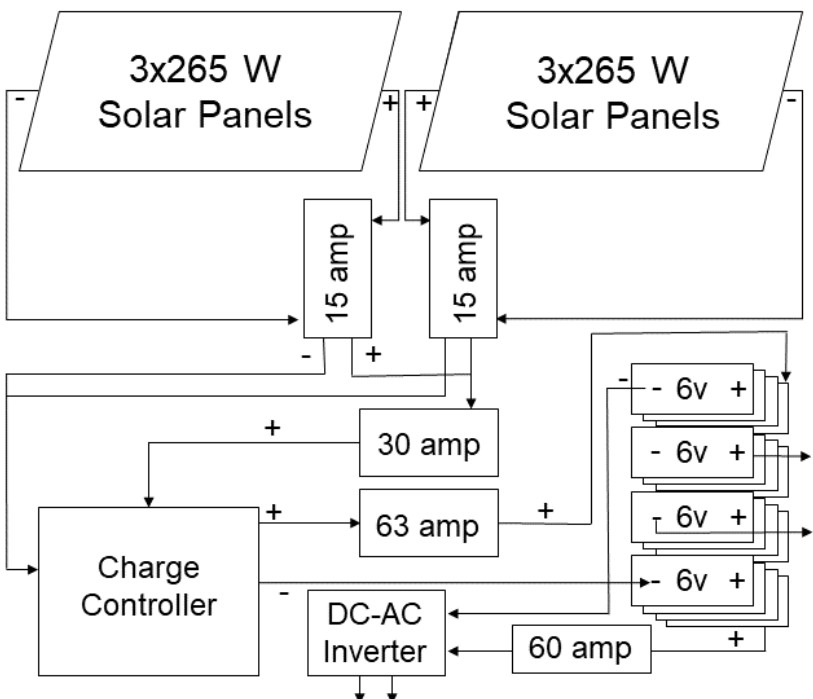

**Fig. 7. Schematic of the solar and power supply system with wiring and circuit breakers. Wires are noted positive (+) and black are negative (-). Arrows from the DC-AC inverter supply 120-volt AC. Arrows from the battery back supply 12-volt DC. Circuit breakers are labelled by their ampere (amp) rating. Batteries for the battery bank are labelled by individual battery voltage.**





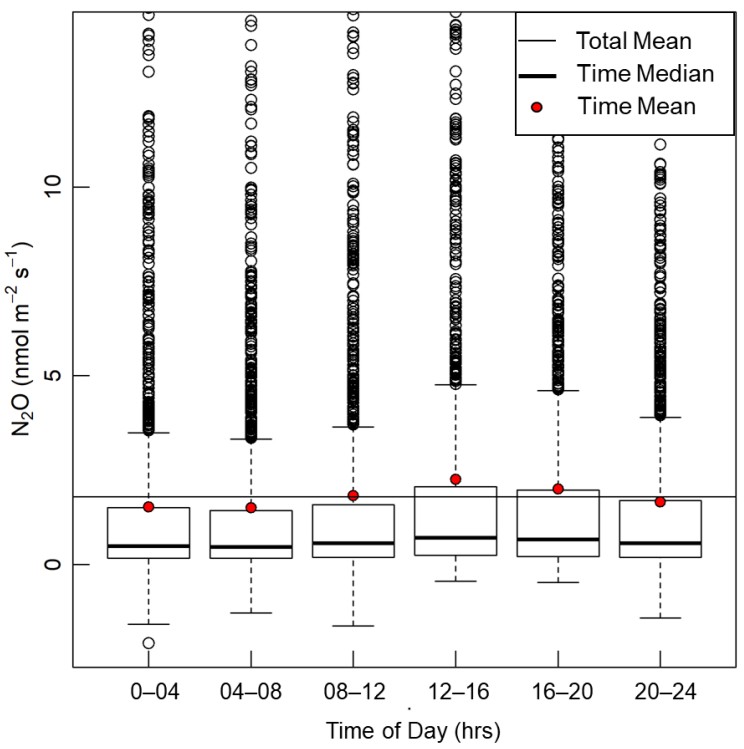

**Fig. 8. Boxplot of N₂O fluxes during each four-hour interval. Positive outliers that comprised 1.9% of the total dataset were greater than 14 nmol m⁻² s⁻¹.**





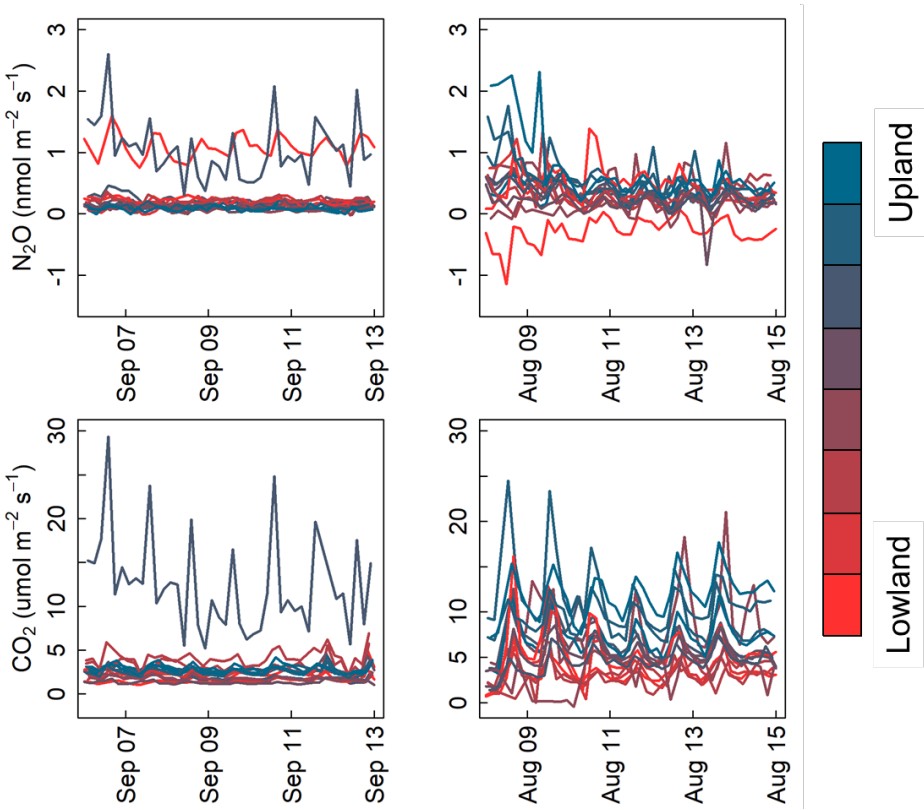

**Fig. 9. $N_2O$ and $CO_2$ flux time series shaded by plot topographic location over two one-week periods in September 2017 and August 2018.**






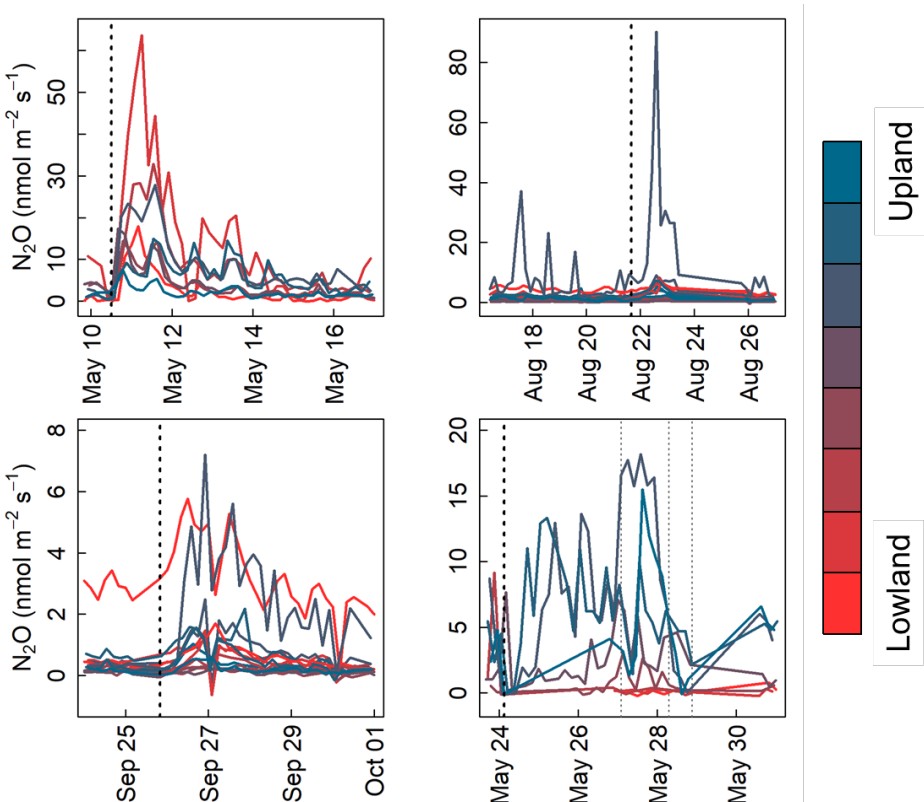

**Fig. 10.** N$_2$O flux time series shaded by plot topographic location over four one-week periods in May 2017, August 2017, September 2017, and May 2019. Black dashed lines denote rain events analyzed for peak delay, grey lines indicate rain events that did not fit our selection criteria and were over 2 mm d$^{-1}$.