# Peer review of "Capturing temporal heterogeneity in soil nitrous oxide fluxes with a robust and low-cost automated chamber apparatus"

_Atmospheric Measurement Techniques, 2020_

## Referee Comment (RC1) · Anonymous Referee #1 · 18 Apr 2020

Interactive comments on "Capturing temporal heterogeneity in soil nitrous oxide fluxes with a robust and low-cost automated chamber apparatus" by Lawrence and Hall Reviewer prefers to be anonymous The authors present here a significant promise in implementing low-cost but robust automated chambers for intensive temporal soilborne GHG flux measurements. The paper describes the details of the hardware of chamber design, chamber operation, measurement principles, troubleshooting, and data to support the sound functioning of the design. Given the high temporal variability, especially for N2O fluxes, high-resolution measurements are critical and often achieved by automated chambers. However, their use has been limited due to the expensive nature of the technology. Therefore, ∼$40,000 USD for 16 automated chambers with the level of accuracy and robustness as shown in this study is a significant development. This could lead to greater adoption of automated chambers to curb the uncertainty of N2O flux estimates. Therefore, I think the paper should be published in AMT. I have listed a few questions and suggestions below for the authors' consideration. 1) I was a little confused about how many chambers were closed at a time. For example, with ∼30 min closure period/chamber, only eight chambers could be measured in a four-hour sampling loop. A bit more clarification could be helpful. Also, how did you program the sequence of chamber closure (chamber #1 to 16) during each sampling loop? Was it random or fixed? This might impact bias. 2) One potential pitfall of automated chambers operating at a sub-daily scale is that they can keep the chamber close for a substantial amount of time in a day that can intercept the rainfall. This can impact soil moisture content inside the chamber relative to outside soils. However, this design reduces the closure period to 30 min (usually 45 min to 1 hour in other designs). With 6 sampling loops (4 hours long each), this could keep the chambers closed for 3 hours a day. I am interested to know if this design can be programmed in such a way to not close the chamber when there is rainfall/precipitation happening to allow the water inside the chamber? 3) A table outlining side-by-side similarities and differences (pros and cons) with other automated systems would be interesting. I understand that the authors have discussed that here and there, but a summary would be helpful.
* * *

---

## Referee Comment (RC2) · Anonymous Referee #2 · 29 Apr 2020

An interesting and useful paper outlining a relatively simple and robust technique for automated field chambers. Only a few minor comments on the operation and design of the chambers themselves, but I'd like to see more discussion around the construction labour costs (important if claiming "low cost" but not including them in the budget) and also the availability/cost of replacement parts - particularly if these are custom built. Some comment on the technical requirements for the data analysis would also be useful for handling such a large dataset.

Other comments: Introduction line 63. Some references to these other measurement

types are required here. Line 65: There are a lot of automated systems that use GC's as well which need to be referenced here. These are also relatively inexpensive (<\$20,000 USD) compared to the lasers and have been used in extreme environments (e.g. Wolf 2010 in Inner Mongolia and Kiese 2003 in tropical rainforests). These need to be mentioned as existing options. Line 125: Clarify that these measurements are referring to the frame and not the "collar". Chamber base and collar are both often used to describe the same thing Line 167: I imagine this would be a major limitation in highly shrink/swell soils such as vertisols, or large vigorous crops (please comment) Line 195: What diamter and material is used for the chamber lines (I may have missed elsewhere)

---

## Author Comment (AC1) · 27 May 2020

We thank the reviewer for their thoughtful comments, which we address below.

For the editor's convenience, the line numbers in our quoted text refer to the revised manuscript version which we will submit.

Anonymous Referee #1 The authors present here a significant promise in implementing low-cost but robust automated chambers for intensive temporal soilborne GHG flux measurements. The paper describes the

details of the hardware of chamber design, chamber operation, measurement princi-
ples, troubleshooting, and data to support the sound functioning of the design. Given
the high temporal variability, especially for N2O fluxes, high-resolution measurements
are critical and often achieved by automated chambers. However, their use has been
limited due to the expensive nature of the technology. Therefore, âĽij$40,000 USD for
16 automated chambers with the level of accuracy and robustness as shown in this
study is a significant development. This could lead to greater adoption of automated
chambers to curb the uncertainty of N2O flux estimates. Therefore, I think the paper
should be published in AMT.

Response: We appreciate the reviewer's interest in our manuscript.

I have listed a few questions and suggestions below for the authors' consideration. 1)
I was a little confused about how many chambers were closed at a time. For example,
with âĽij30 min closure period/chamber, only eight chambers could be measured in a
four-hour sampling loop. A bit more clarification could be helpful. Also, how did you
program the sequence of chamber closure (chamber #1 to 16) during each sampling
loop? Was it random or fixed? This might impact bias.

Response: Good point. Section 2.5 "Measurement Principle" has been amended to
clarify chamber closure with 16 chambers and chamber measurement sequence.

L252: "When sixteen chambers were deployed, a new chamber was closed every fif-
teen minutes and two chambers were closed simultaneously with the sample gases
vented during a 15-minute equilibration period prior to a 15-minute measurement pe-
riod. Here we describe the eight-chamber arrangement. To reduce possible conflation
between measurement time and plot topographic position, we chose a consistent but
staggered measurement sequence for each four-hour period (1, 5, 3, 7, 2, 6, 4, 8),
where plot one was the lowest topographic position. When sixteen chambers were
deployed, the plot sequence was maintained so paired chambers at each plot were
measured in a single half-hour cycle."

We also added text in section 2.4 "Principles of Gas Sampling" to briefly explain the mechanics of how the chamber sample selection was modified to accommodate more chambers.

L236: "To operate sixteen chambers without reducing measurement period or frequency, separate parallel selection manifolds, additional mass flow controllers for chamber inlet/outlet, and diaphragm pumps were added. Two additional solenoid valves on the sample selection manifold allowed selection between each of the two inlet and outlet manifolds."

2) One potential pitfall of automated chambers operating at a sub-daily scale is that they can keep the chamber close for a substantial amount of time in a day that can intercept the rainfall. This can impact soil moisture content inside the chamber relative to outside soils. However, this design reduces the closure period to 30 min (usually 45 min to 1 hour in other designs). With 6 sampling loops (4 hours long each), this could keep the chambers closed for 3 hours a day. I am interested to know if this design can be programmed in such a way to not close the chamber when there is rainfall/precipitation happening to allow the water inside the chamber?

Response: This is a good point that bears addressing in further detail in subsequent work. Potential impacts of chambers on soil moisture are one limitation of any chamber method. In principle, a voltage signal from a rain gauge could be easily programmed to signal the chambers to remain open during rainfall events as implemented by Butterbach-Bahl and Dannenmann (2011). We are providing our datalogger code associated with this paper in a public repository at Iowa State University (doi to be assigned following manuscript acceptance), which illustrates the method by which chamber movement is controlled and could be modified. However, there are logistical complications of this approach. In our region, prolonged low-intensity rainfall events are common, and this could result in long periods of time (many hours to days) without any measurements. Including a rainfall rate threshold (i.e. >0.25 cm in a measurement period) required to open chambers could limit the frequency and duration of data gaps.

[Figure]

Reducing the measurement frequency or measurement period would also limit the proportion of time that the chambers are closed. Both solutions would limit the amount of data collected. We plan on quantifying the magnitude of any soil moisture effect of chamber closure in our ongoing work. A description of this issue has been added to section 3.1 "Troubleshooting" including a citation to a more detailed discussion of the issue.

L351: "During periods of chamber closure (3 out of every 24 hours during typical operation), rainfall was excluded from the chamber enclosure, which could potentially alter soil moisture. Elsewhere, a rain gauge has been used to signal automated chambers to remain open during rainfall events (Butterbach-Bahl and Dannenmann, 2011). Here, we elected to maintain a consistent measurement schedule irrespective of rainfall, due to the logistical challenges posed by prolonged rainfall events (when no measurements would be collected). A rainfall rate threshold to open the automated chambers could limit the frequency and duration of data gaps in future studies.. Future measurements will quantify the potential magnitude of any soil moisture effect associated with our auto-chamber system. To reduce the duration that the chambers were closed when the system was off for power conservation or maintenance, we either left the compressor on and the chambers in the open position, or propped the chambers open."

3) A table outlining side-by-side similarities and differences (pros and cons) with other automated systems would be interesting. I understand that the authors have discussed that here and there, but a summary would be helpful.

Response: Good point. Text has been significantly expanded in section 1 "Introduction" (lines 61-81) summarizing the benefits and limitations of various analyzer/chamber options in field settings, along with citations describing these approaches. We did not provide an explicit table in the manuscript because this issue has been addressed in detail elsewhere (e.g. Fassbinder et al. 2013), and because of the difficulty of categorizing the diversity of commercial and custom-built automated chamber methodologies. We further elaborate on the basic principles of static and dynamic chamber operation

(pros/cons) on lines 99-116.

L61: "Prefabricated automated chambers capable of measuring soil trace gas fluxes are available commercially and can be plumbed to a wide range of analyzers—-most commonly, infrared gas analyzers that measure CO2. Commercially available chambers typically rely on electric components for movement which are sensitive to moisture, and they are substantially more expensive (often many thousands of USD) than the chamber design described here (materials costs of ∼500 USD/chamber). Other custom-built chamber designs have been developed to address specific research needs (Ambus and Robertson 1998; Butterbach-Bahl et al., 1997; Savage et al., 2014). Chambers have been paired with analyzers to measure other trace gases, including N2O and CH4, by utilizing methods such as gas chromatography (GC), photo-acoustic infrared detection, tunable diode laser (TDL), or cavity ring-down laser spectroscopy (Ambus and Robertson, 1998; Breuer et al., 2000; Courtois et al., 2019; Papen and Butterbach-Bahl, 1999; Pihlatie et al., 2005). Fassbinder et al. (2013) provide a detailed summary of the advantages and limitations of each analyzer option that we briefly summarize here. GC systems equipped with electron capture detectors (ECD) have been used to measure N2O from automated chambers (Breuer et al., 2000; Papen and Butterbach-Bahl, 1999). However, GC systems have high power demand and require carrier gases and radioactive elements for ECD operation that may limit their field practicality. Interference by water vapor potentially limits the use of photoacoustic analyzers in the field (Ambus and Robertson, 1998; Fassbinder et al., 2013). Laser-based analytical approaches are capable of rapid (e.g. 10 Hz) and precise N2ňO measurements, but these analyzers may be prohibitively expensive (>70,000 USD) and also have relatively high power requirements for autonomous field deployment (Fassbinder et al., 2013; Pihlatie et al., 2005). We sought to implement a lower-cost, solar powered, soil gas flux measurement system capable of operating unattended in a harsh field environment, and where analyzers could feasibly be replaced if stolen or damaged. For these reasons, we utilized a gas filter correlation (GFC) infrared N2O analyzer in our study (∼16,000 USD), similar to that described previously by Fassbinder et al. (2013),

along with an infrared gas analyzer for CO2/H2O measurement (∼4,000 USD). However, other analyzers could be readily employed with the chamber and manifold system described below."

---

## Author Comment (AC2) · 27 May 2020

We thank the reviewer for their thoughtful comments, which we address below. For the editor's convenience, the line numbers in our quoted text refer to the revised manuscript version which we will submit

Anonymous Referee #2 An interesting and useful paper outlining a relatively simple and robust technique for automated field chambers. Only a few minor comments on the operation and design of the chambers themselves, but I'd like to see more discussion around the construction labour costs (important if claiming "low cost" but not including them in the budget) and also the availability/cost of replacement parts - particularly if these are custom built.

Response: Good point. We had noted the manufacturers of the major components in the original manuscript but we have now presented all of this information more explicitly in a table of chamber component cost, supplier, and use which has been added as Appendix A. We have also included estimates of labor hours for construction/assembly (labor costs would vary greatly depending on the wage of the person doing the work) along with a more detailed enumeration of materials costs.

L473: "Despite these challenges, we were able to construct and maintain 8 (+1 spare) high-frequency automated chambers for sub-daily N2O and CO2 flux measurements in a temperate agricultural field, with a total materials cost (∼$40,000 US dollars, including parts for 9 chambers, gas analyzers, control system, and power supply) that is a fraction of the cost of many laser-based N2O analyzers alone. We estimate that the chambers and control system took us 130–260 hours in total to construct and troubleshoot (with concomitant labor/salary costs) and did not require specialized tools beyond those available in a typical workshop."

Some comment on the technical requirements for the data analysis would also be useful for handling such a large dataset.

Response: Good Point. We amended section 2.5 "Measurement Principle" to describe ancillary data files that will accompany the final manuscript to describe these analyses. We are providing our datalogger and analysis code associated with this paper in a public repository at Iowa State University (doi to be assigned following manuscript acceptance).

L292: "All data cleaning, flux calculation, and data analysis were conducted with R statistical software version 3.6.1 (R Core Team, 2019). Cleaning and calibration required R packages lubridate, nlme, and reshape (Spinu, 2020; Wickham, 2018; Willigen, 2020). The CR3000 datalogger code we used to operate the chambers and record data, along with an example dataset and R script for data cleaning and flux calculations, are provided as archived files associated with this publication."

Other comments: Introduction line 63. Some references to these other measurement types are required here.

Response: We have amended this to include reference to more measurement types, including GC systems.

L66: "Chambers have been paired with analyzers to measure other trace gases, including N2O and CH4, by utilizing methods such as gas chromatography (GC), photoacoustic infrared detection, tunable diode laser (TDL), or cavity ring-down laser spectroscopy (Ambus and Robertson, 1998; Breuer et al., 2000; Courtois et al., 2019; Papen and Butterbach-Bahl, 1999; Pihlatie et al., 2005)."

Line 65: There are a lot of automated systems that use GC's as well which need to be referenced here. These are also relatively inexpensive ((<$20,000 USD) compared to the lasers and have been used in extreme environments (e.g. Wolf 2010 in Inner Mongolia and Kiese 2003 in tropical rainforests). These need to be mentioned as existing options.

This is a relevant point, but we note that whereas chamber systems themselves may be < $20,000, we are unaware of any modern gas chromatographs (with electron capture detector for N2O analysis) that can themselves be purchased for less than many tens of thousands of dollars. We have added GC measurement systems (as described above) and cited additional studies to demonstrate applications of each analyzer type. We also expanded a paragraph in the Introduction to cover more analyzer options and details.

L61: "Prefabricated automated chambers capable of measuring soil trace gas fluxes are available commercially and can be plumbed to a wide range of analyzers—-

most commonly, infrared gas analyzers that measure CO2. Commercially available chambers typically rely on electric components for movement which are sensitive to moisture, and they are substantially more expensive (often many thousands of USD) than the chamber design described here (materials costs of ∼500 USD/chamber). Other custom-built chamber designs have been developed to address specific research needs (Ambus and Robertson 1998; Butterbach-Bahl et al., 1997; Savage et al., 2014). Chambers have been paired with analyzers to measure other trace gases, including N2O and CH4, by utilizing methods such as gas chromatography (GC), photo-acoustic infrared detection, tunable diode laser (TDL), or cavity ring-down laser spectroscopy (Ambus and Robertson, 1998; Breuer et al., 2000; Courtois et al., 2019; Papen and Butterbach-Bahl, 1999; Pihlatie et al., 2005). Fassbinder et al. (2013) provide a detailed summary of the advantages and limitations of each analyzer option that we briefly summarize here. GC systems equipped with electron capture detectors (ECD) have been used to measure N2O from automated chambers (Breuer et al., 2000; Papen and Butterbach-Bahl, 1999). However, GC systems have high power demand and require carrier gases and radioactive elements for ECD operation that may limit their field practicality. Interference by water vapor potentially limits the use of photoacoustic analyzers in the field (Ambus and Robertson, 1998; Fassbinder et al., 2013). Laser-based analytical approaches such are capable of rapid (e.g. 10 Hz) and precise N2ǑO measurements, but these analyzers may be prohibitively expensive (>70,000 USD) and also have relatively high power requirements for autonomous field deployment (Fassbinder et al., 2013; Pihlatie et al., 2005). We sought to implement a lower-cost, solar powered, soil gas flux measurement system capable of operating unattended in a harsh field environment, and where analyzers could feasibly be replaced if stolen or damaged. For these reasons, we utilized a gas filter correlation (GFC) infrared N2O analyzer in our study (∼16,000 USD), similar to that described previously by Fassbinder et al. (2013), along with an infrared gas analyzer for CO2/H2O measurement (∼4,000 USD). However, other analyzers could be readily employed with the chamber and manifold system described below."

Line 125: Clarify that these measurements are referring to the frame and not the "collar". Chamber base and collar are both often used to describe the same thing

Response: Good point. In the context of this paper, frame refers to the stainless-steel tubing while "base" refers to the plastic structure that the frame is attached to. This has been clarified in section 2.2 "Chamber Design".

L147: "Here we define the chamber base as the rigid, rectangular polyethylene structure (Figure 3a) and the chamber frame as the metal structure superior to the base which allows for movement of the chamber lid (Figure 3). The chamber collar is defined as the length of polyvinylchloride (PVC) pipe that forms the interface between the chamber lid and the soil."

Line 167: I imagine this would be a major limitation in highly shrink/swell soils such as vertisols, or large vigorous crops (please comment)

We now clarify that these soils did in fact contain swelling clays, albeit not to the extent of a true Vertisol. We found that pounding rebar into the soil on either side of the chamber base and affixing the frame to the rebar (described in the text below that referenced in this comment) addressed problems associated with chamber movement. This solution is noted in section 2.2 "Chamber Design". Application of this method in true Vertisols could likely be achieved by deeper installation of rebar to secure the chamber. We periodically checked that the chamber lids were effectively sealing against the collars. See clarified text:

L190: "However, we found that pressure exerted by the pneumatic arm when opening or closing the chamber occasionally shifted the position of the chamber base or collar and prevented a seal between the chamber lid, collar, and soil. This occasionally occurred following tillage or when soils were extremely dry, given that these soils contained swelling clays. To address this problem, we anchored the chamber base using two steel rebar rods (60 cm length, 1.27 cm diameter) pounded 45 cm into the ground on either side of the chamber base and affixed to the outside of the chamber base

with U-bolts positioned along the central axis of the collar (Fig. 3). We periodically checked that the chamber lids were effectively sealing against the collars. Application of this method to true Vertisols, with even greater shrink/swell behavior, could likely be achieved using similar use of rebar to anchor the chamber."

We acknowledge that vegetation can be a challenge for chamber-based measurements. We added details about how we dealt with vegetation management in section 2.1 "Study Site"

L131: "Chambers were placed immediately adjacent to crop plants; due to frequent tillage and herbicide application, recruitment of other plants inside the chamber collars was uncommon, but any plants were removed from the chamber interiors as soon as they were observed. Roots from crop plants were not excluded and likely grew beneath chambers."

Line 195: What diamter and material is used for the chamber lines (I may have missed elsewhere)

Response: This was unclear, good point. The pressurized tubing details are noted in section 2.3 "Chamber Lid Operation". The chamber lines are the same material, we have added the material details to section 2.4 "Principles of chamber gas sampling" as well to make that more clear.

L200: "We used 0.64 cm OD, 0.43 cm ID low-density polyethylene (LDPE) plastic tubing. We initially used aluminum composite tubing (Synflex 1300), which has been commonly used in other field trace gas measurement studies (e.g. Bowling et al. 2015), but we found this to be impractical for our application given its vulnerability to kinking during chamber installation and removal through dense vegetation."
* * *